ecology, environmental science

succession, hydrothermal vent, disturbance, resilience, seafloor eruption, colonization

**Author for correspondence:**
L. S. Mullineaux
e-mail: lmullineaux@whoi.edu

# Prolonged recovery time after eruptive disturbance of a deep-sea hydrothermal vent community

L. S. Mullineaux[1], S. W. Mills[1], N. Le Bris[2], S. E. Beaulieu[1], S. M. Sievert[1] and L. N. Dykman[1]

[1]Biology Department, Woods Hole Oceanographic Institution, Woods Hole, MA, USA
[2]CNRS-Sorbonne Université, Benthic Ecogeochemistry Laboratory, Banyuls-sur-Mer, France

LSM, 0000-0002-8932-2625; SWM, 0000-0001-6266-6092; NLB, 0000-0002-0142-4847; SEB, 0000-0002-2609-5453; SMS, 0000-0002-9541-2707; LND, 0000-0002-4030-7897

Deep-sea hydrothermal vents are associated with seafloor tectonic and magmatic activity, and the communities living there are subject to disturbance. Eruptions can be frequent and catastrophic, raising questions about how these communities persist and maintain regional biodiversity. Prior studies of frequently disturbed vents have led to suggestions that faunal recovery can occur within 2–4 years. We use an unprecedented long-term (11-year) series of colonization data following a catastrophic 2006 seafloor eruption on the East Pacific Rise to show that faunal successional changes continue beyond a decade following the disturbance. Species composition at nine months post-eruption was conspicuously different than the pre-eruption 'baseline' state, which had been characterized in 1998 (85 months after disturbance by the previous 1991 eruption). By 96 months post-eruption, species composition was approaching the pre-eruption state, but continued to change up through to the end of our measurements at 135 months, indicating that the 'baseline' state was not a climax community. The strong variation observed in species composition across environmental gradients and successional stages highlights the importance of long-term, distributed sampling in order to understand the consequences of disturbance for maintenance of a diverse regional species pool. This perspective is critical for characterizing the resilience of vent species to both natural disturbance and human impacts such as deep-sea mining.

## 1. Introduction

Deep-sea hydrothermal vents host intriguing communities that are fuelled by chemosynthesis and occur on active seafloor geological features such as mid-ocean ridges, back-arc basins, and seamounts. The habitat in these settings is patchy and subject to catastrophic eruptive disturbance, so the persistence of vent species depends on their ability, predominantly as larvae, to disperse and colonize other vents. In order to understand the resilience of this system, it is important to quantify how quickly vent communities recover from disturbance, what processes influence their ability to recover, and how processes at an individual vent field affect community dynamics, persistence, and species diversity on the regional scale.

The ability of a species to disperse successfully from one vent patch to another depends on factors including the spatial separation of the vents, larval dispersal capabilities, and habitat requirements of new colonists [1]. Vents occur in patterns aligned with locations of magma intrusion into the ocean crust at plate boundaries, often along topographic features such as mid-ocean ridges, volcanic arcs, and back-arc spreading centres [2]. In regions with relatively high magma supply rate, such as the East Pacific Rise (EPR), vent fields are separated by a few to tens of kilometres [3], whereas in areas with slower spreading rates, such as the Mid-Atlantic Ridge

(MAR) and some back-arc spreading systems, vent fields may be separated by hundreds of kilometres [2]. Larvae are transported in ocean currents, and must locate suitable physical and chemical habitat within a vent field to colonize successfully. Vent fluid chemistry varies across individual vent orifices and over time [4]. Species differ in their vent habitat requirements, so colonization of any one species depends not only on its ability to reach a new vent but on the habitat characteristics it encounters there. Even if larvae colonize successfully, a population may fail to persist as the local conditions change, or as it interacts with other species during succession.

Disturbances at vent-field scale that eradicate the local community occur mainly through volcanic eruptions or landslides. The frequency of disturbance may range from almost continuous at vents on submarine arc volcanoes with multiyear eruptions (e.g. NW Rota-1; [5]) to very infrequent at vents on slow-spreading ridges. At spreading ridges, the frequency of volcanic eruptions (like the spacing of vents) is related to magma supply: at the fast-spreading EPR, vent communities are eradicated by seafloor eruptions on decadal timescales [6], whereas at the slow-spreading MAR an eruption might not occur for 10 000 years [7]. These natural eruptive disturbances appear to be a fundamental regulating feature of vent populations and communities at volcanically active areas, as both the disturbance frequency and vent spacing are expected to influence connectivity among vents [8].

A fundamental question in vent ecology is the resilience of communities to disturbance, including the recovery time in an individual vent field, and the regional persistence of groups of communities (metacommunities) connected across fields by dispersal. Efforts to investigate recovery of vent communities, by monitoring colonization and species assembly, have taken advantage of local extinctions driven by seafloor volcanic eruptions at vents on the EPR [9–11] and Juan de Fuca Ridge [12,13]. These field studies, all conducted in frequently disturbed systems, have been interpreted to suggest that typical vent community recovery times may be as short as 2–4 years [14], and are characteristic of a resilient fauna adapted to disturbance events [15]. The caveats on those estimates, and assumptions underlying the interpretations, need to be examined critically. Identifying a pre-disturbance baseline can be difficult, and the time to re-establish a disturbed state may be much shorter than to establish a climax community, especially if the regional diversity is high (figure 1). Furthermore, recovery of vent communities in other regions exposed to much lower frequencies of catastrophic disturbance, such as the MAR or some regions of the western Pacific, may be slower [8,16]. Communities in those regions appear to be relatively stable over periods of years to decades, in the absence of disturbance on a vent-field scale [17,18]. Understanding the processes contributing to recovery of both community structure and function is essential for interpreting these field results and using them to predict resilience more generally across a spectrum of disturbance regimes.

Renewed interest in extracting resources from deep-sea mineral deposits [19] has led to an urgent need for a better understanding of vent community recovery and metacommunity resilience. The prospect of seafloor mining of sulfide deposits puts vent communities at risk in some of the areas where connectivity and resilience are least well known. The first demonstration of sulfide mining at a deep-sea vent occurred in the Okinawa Trough in 2017. As of 2019, eight nations had polymetalic sulfide mining exploration contracts under the International Seabed Authority, and another eight

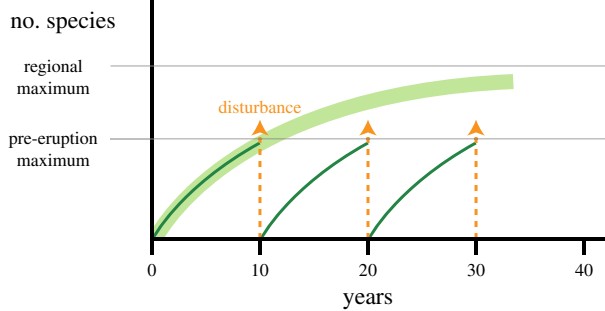

**Figure 1.** Conceptual framework for community recovery in frequently disturbed deep-sea hydrothermal vent systems. Increase in number of species following frequent disturbance events (thin curves), does not reach potential species number in a community subject to less frequent disturbance (thick curve). Pre-eruption species number is not a consistent baseline because it depends on the time since prior disturbance. (Online version in colour.)

had allowed mining research or exploration at vents in waters of their national jurisdiction [20].

The specific objectives of this study are to investigate the rate and nature of ecological succession after a catastrophic disturbance. We take advantage of a massive seafloor eruption in 2006 on the EPR where we were able to initiate post-eruption observations within months and continue them uninterrupted for more than a decade, providing an extensive colonization time series that is unprecedented for the deep sea. These data enable a critical evaluation of recovery time in a frequently disturbed vent system, and its relevance to a broader spectrum of disturbance regimes. The vent system represents an intriguing case for succession, where both connectivity and disturbance are closely tied to earth dynamics, and informs broader efforts to understand and predict changes in community structure in response to natural and human disturbance.

## 2. Material and methods

Experiments and habitat characterization for this study were conducted on the EPR, on a series of cruises starting five months after a catastrophic eruption in January 2006 that eradicated the communities in a vent field located along the ridge between 9°48′ and 9°52′ [21]. Vent communities at 9°47′ N and further south persisted. A prior eruption in 1991 had also eradicated communities in this part of this ridge segment [9]. We had been monitoring larval supply and colonization at these vents in the years between the two eruptions, and participated in a rapid-response cruise directly after the 2006 event. In the 11 years since, we have accumulated a near-continuous record of colonization at sites at 9°50′ N (figure 2a) and neighbouring vents, in some cases with associated measurements of larval supply and/or vent fluid composition.

Colonization monitoring was focused at a diffuse-flow vent site near P-vent (9.838° N, 104.291° W, 2509 m depth) where venting continued through the duration of the study. Colonization surfaces were deployed and recovered using the human-occupied submersibles Alvin and Nautile, and the remotely operated vehicle Jason, with recoveries occurring at time points of 9, 11, 22, 33, 96, 106, and 135 months after the eruption (figure 2a). A few early (nine months) samples were included from a transient vent near the Ty/Io vent, 200 m south of P-vent, but that site was abandoned when it became clear that venting had waned precipitously after the first cruise.

Colonists (larvae that had settled) were collected on experimental surfaces, termed 'sandwiches', comprising six layered $10 \times 10$ cm Lexan plastic plates, each 0.7 cm thick and separated from each other by 1 cm to provide roughly 12 000 cm$^2$ surface

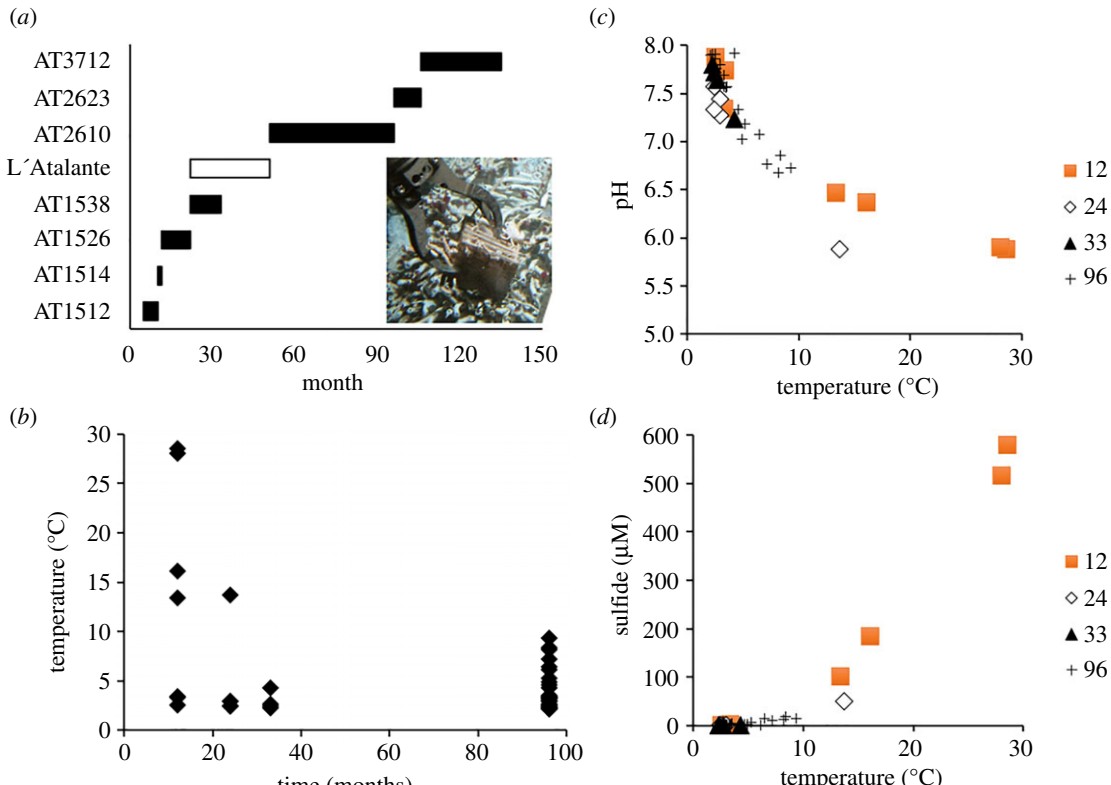

**Figure 2.** Experimental design and biologically relevant chemical characteristics at P-vent near 9° 50' N on the East Pacific Rise. (*a*) Schedule of deployment and recovery of colonization sandwiches on research vessels Atlantis (AT designates cruise number) and L'Atalante, in months after the 2006 eruption (those recovered on the L'Atalante cruise were unusable due to ferrous precipitate; inset shows colonized sandwich in Alvin manipulator; (*b*) change in temperature over months post-eruption; (*c*) relationship of pH with temperature; (*d*) relationship of total sulfide with temperature. (Online version in colour.)

area (figure 2*a*). For each deployment, replicate sandwiches were placed in each of three distinct habitat types, characterized by temperature as hot (7–30°C), warm (3–7°C), or cool (2–3°C). These habitat types correspond to the faunal zones listed by Micheli *et al.* [22] as vestimentiferan (tubeworm), bivalve, and suspension-feeder, respectively. Habitat temperature was measured at the base of each sandwich, on deployment and recovery, with a temperature probe for 1–2 min until a clear maximum value was observed. The maximum value of the recovery temperature was used to characterize the thermal environment of the surface, for consistency with prior studies. These temperature measurements are useful for characterizing the general thermal habitat of a sandwich, even though the specific temperature experienced by an individual colonist could vary over the extent of the surface and over tidal and shorter timescales. Due to logistical limitations, not all habitats were sampled for each interval. In many cases, the thermal habitat changed over the course of a deployment, as temperatures overall tended to decrease with time (figure 2*b*). To account for this variation, both the deployment zone and the recovery temperature were taken into account when interpreting colonization results.

On recovery, each sandwich was placed in an individual collection compartment for transport back to the ship and subsequent analysis of colonists. On board, sandwiches and their attached colonists were preserved in 80% ethanol, as were any individuals that had become detached in the compartment and were retained on a 63 μm sieve. In the laboratory, each surface was examined under a dissecting microscope and all metazoan colonists (including detached individuals greater than 1 mm) were enumerated and identified to species or the lowest taxonomic group possible. Meiofauna that were too small or mobile to be retained reliably in the samples, or outside our taxonomic expertise, were not included in the analyses.

Species or morphogroup abundance was assessed for each recovery date on three replicate sandwiches from each available

habitat. Abundance of a subset of species, selected based on their demonstrated role in prior succession studies [9,23] and/or prominence in the current samples, was compared across time with a non-parametric Kruskal–Wallace test (Systat v. 13). Analyses of the full community species composition were conducted with nonmetric multidimensional scaling (nMDS using Pearson correlation matrix; Systat v. 13) to visualize the change in composition over time. Relative abundance (compared to total number of individuals in the sample) was used in the nMDS to compensate for different deployment intervals or surface types, and log transformed to avoid dominance of pattern by abundant species. Numbers of colonists varied widely between samples, so diversity was assessed by a rarefaction analysis (BioDiversity Pro), with total individuals pooled over replicates. Single-celled protozoans were not included in nMDS or rarefaction analyses, as their occasional outlier abundances obscured patterns displayed by the metazoans.

Colonization data from prior to the eruption were available from several vents located within a few hundred metres of the P-vent site [24], and were selected from the East Wall vent collections to represent a range of thermal habitats similar to P-vent, an extended deployment time (37 months), and the most mature successional stage available (85 months after the previous eruption in 1991). These pre-eruption colonization experiments had been conducted on basalt blocks with similar collecting characteristics but smaller surface area than the sandwiches [25], so comparisons with post-eruption results were made on relative abundance.

Free sulfide (i.e. $S^{2-} + HS^- + H_2S$), pH, and temperature were measured in close vicinity to colonizers at 11, 24, 33, and 96 months after the 2006 eruption as in [25]. We used an $Ag/Ag_2S$ sulfide electrode (0.8 mm diameter) associated with a miniaturized glass electrode (1.5 mm) connected to individual potentiometric loggers (NKE SPHT) at 11, 22, and 33 months [26]. A 0.1 mm-disc mercury amalgam electrode connected to an underwater

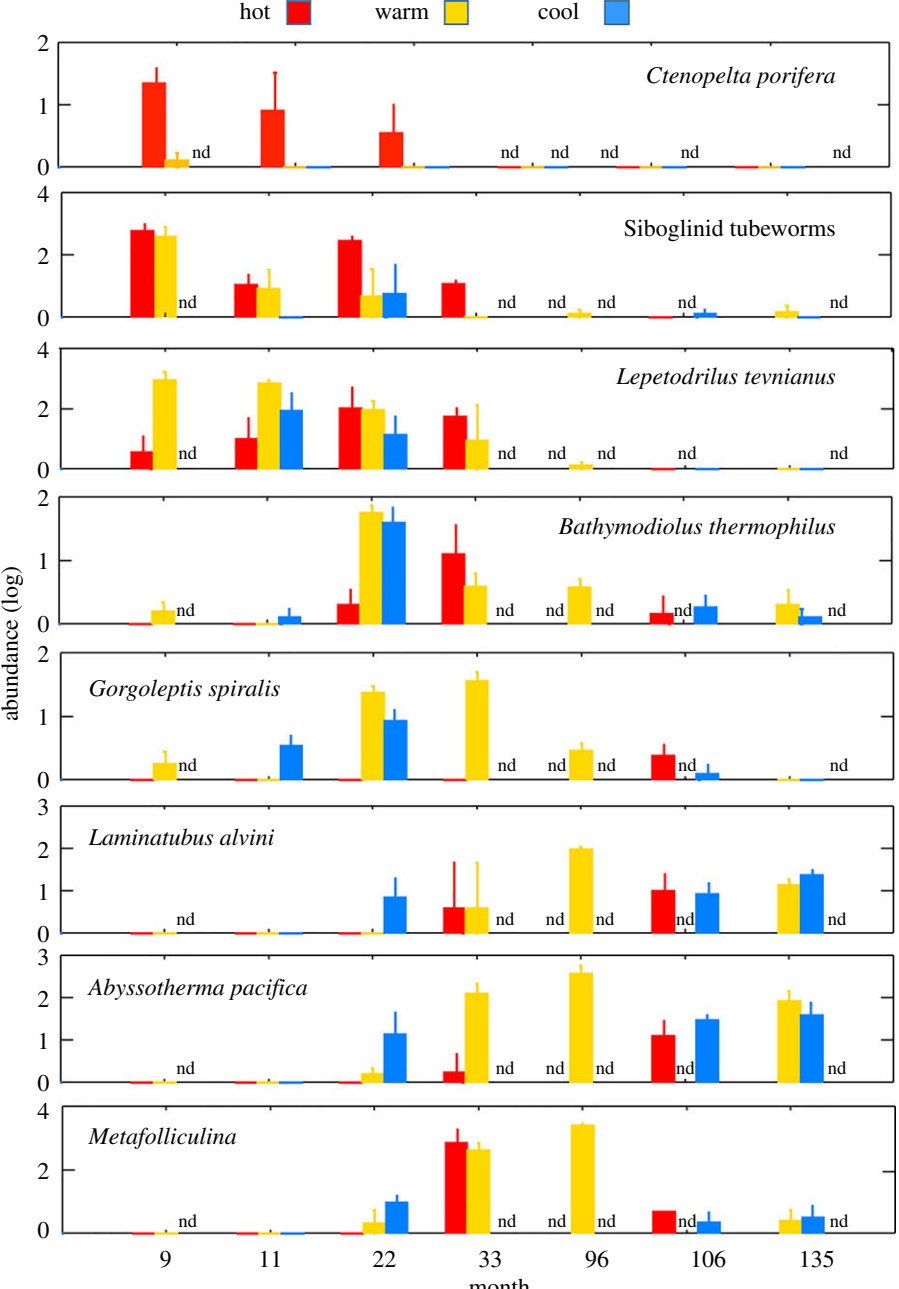

**Figure 3.** Abundance (log-transformed) of prominent species colonizing sandwiches at P-vent after the 2006 eruption. Values are mean of replicates (usually 3) deployed in hot, warm, and cool habitat. Bars are standard error; 'nd' indicates no sandwiches recovered.

potentiostat (NKE SPOT-L) was used to measure free sulfide by cyclic voltammetry at month 96. The electrodes were combined with a temperature probe (3 mm diameter, NKE S2T6000). Precise positioning and stabilization of the tip using the submersible arm allowed continuous data records of 1 to 3 min on each location (at the frequency of 4–12 measurements per minute for potentiometry and one triplicate measurement per minute for voltammetry). Average values defined for each measurement sequence integrate fluctuations and allow characterizing chemical gradients as a function of temperature with a spatial resolution much finer than the sandwich size.

Abiotic conditions around sandwiches at P-vent after the eruption show a typical increase of free sulfide and decrease of pH with temperature (figure 2c,d) consistent with mixing of local diffuse vent fluids with ambient seawater being the primary driver of habitat abiotic ranges. When compared with the first time point (12 months post-eruption), the narrower ranges of sulfide, pH, and temperature at 24, 33, and 96 months indicate habitat conditions with lower vent fluid contribution, depleted in sulfide and less acidic. Temperatures recorded at 24 and 33

months were typically low (figure 2b) because vent fluid flux had decreased at the location of sandwich deployments. Temperatures at 96 months were higher because monitoring had moved to areas with more vigorous venting.

## 3. Results

Colonization of prominent species varied across the series of deployments, showing a distinct progression over time (figure 3; electronic supplementary material, Data Tables S1 and S2). The limpet *Ctenopelta porifera* appeared only in the first 22 months, and primarily in hot habitat. This species may have failed to persist because its preferred high-temperature habitat (typically greater than 15°C; electronic supplementary material, Data Table S1) lasted only 2 to 3 years after the eruption. Its arrival is of particular interest because it was a pioneer from afar [10] and had not been reported within 350 km of this site prior to the 2006 eruption. Siboglinid (vestimentiferan)

Proc. R. Soc. B 287: 20202070

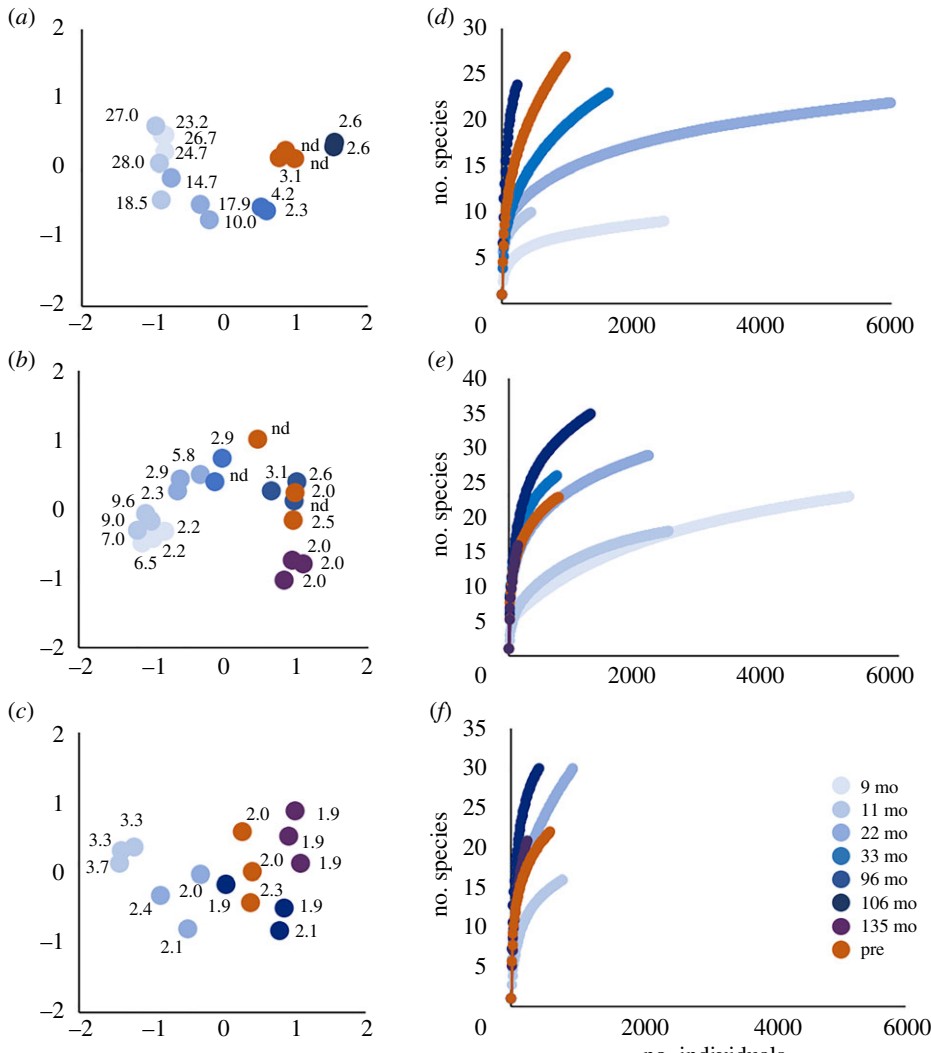

**Figure 4.** Species composition and diversity of colonists at P-vent after the 2006 eruption. Nonmetric multidimensional scaling (nMDS) analyses in hot (*a*), warm (*b*), and cool (*c*) habitat (stress = 0.06, 0.09, 0.10, respectively, where less than 0.05 indicates good fit; proportion of variance = 0.98, 0.96, and 0.94). P-vent samples (blue) are labelled by temperature (°C) and shaded by months post-eruption; pre-eruption samples (orange) provided for comparison. Species assembly (rarefaction) curves in hot (*d*), warm (*e*), and cool (*f*) habitat; data for each recovery date are pooled from replicate samples (usually 3).

tubeworms, including individuals identified as *Riftia pachyptila* and *Tevnia jerichonana*, and other individuals too small to identify to species, arrived predominantly in the first 33 months post-eruption, mostly in hot or warm habitat. They were represented by a few individuals in the 96, 106, and 135 month samples, including some in cool habitat. The limpet *Lepetodrilus tevnianus* also colonized mostly in hot and warm habitat and declined after 33 months. The vent mussel *Bathymodiolus thermophilus* occurred in all habitat types, became abundant at 22 and 33 months after the eruption, and persisted through 135 months. The limpet *Gorgoleptis spiralis* showed a similar temporal pattern, but was found in hot habitat only at 106 months. The small, tube-building polychaete species *Laminatubus alvini* arrived at 22 months, persisted through 135 months, and occurred in all habitats. The protozoans *Abyssotherma pacifica* (a chambered foraminiferan), and *Metafolliculina* sp. (a tube-dwelling ciliate) also occurred at 22 months and beyond in all habitats.

A distinct, directional trend in community composition was evident over the months following the 2006 eruption (figure 4*a*–*c*; electronic supplementary material, Data Table S3). In each of the hot, warm, and cool habitats, composition of the post-eruption samples became progressively more similar to pre-eruption samples during the first 96

months. Later in the series, however, composition bypassed or deviated from the pre-eruption state. Although the pre-eruption samples were intended to represent a 'baseline' community, it is important to remember that they had been collected only 85 months after the previous eruption in 1991. The observed changes in post-eruption composition were directional along one dimension (horizontal axis), but not the other. In some cases, the differences between replicate samples at one recovery date were equal to, or greater than, differences between dates. Both of these patterns indicate that factors other than time influence composition, and microhabitat variation is a likely candidate, as recovery temperatures of the sandwiches varied across replicates.

Species richness of post-eruption colonists, as measured by rarefaction, generally increased over time (figure 4*d*–*f*). For many recovery dates, sample sizes were too low to reach an accurate assessment of full richness (as displayed by lack of asymptote in the curve), but the species number at a given sample size allows for comparison across dates. In the hot habitat, post-eruption richness increased from 9 to 33 months after the eruption, and at 106 months exceeded the pre-eruption level (figure 4*d*). By contrast, in the warm and cool habitat, post-eruption richness reached or exceeded the pre-eruption level by 22 months (figure 4*e*,*f*).

A compilation of vent species reported from the area near 9° N EPR [27] can be added to our observations to identify the potential species pool for recolonization of disturbed vents. Filtering this list so it includes only taxa that could be sampled with sandwiches results in a total of 91 taxa (electronic supplementary material, Data Table S4). Of these, 67 were observed in the post-eruption samples overall, but only 29 were in the 135-month samples, representing 74% and 32% of the regional species pool, respectively.

# 4. Discussion

More than a decade after an eruption destroyed vent communities on the EPR, the species composition of new colonists appeared to still be changing and the successional stage appeared to be different than in the prior, pre-eruption, community. These results suggest that important successional processes continue well beyond the 3–4 years that have previously been monitored after a seafloor disturbance event, and that prior estimates of recovery time need to be re-evaluated.

Temporal changes in environmental conditions clearly influenced succession in this system, especially early in the sequence. In the months directly following the eruption, rapid changes occurred in vent fluid flow distribution, resulting in closure of some newly opened vents [28], and rapid changes in thermal and chemical habitat, as observed at P-vent. We suspect that the rapid decrease of the gastropod *Ctenopelta porifera* and other species during this stage may have been driven by loss of the unique habitat created by the unusually vigorous venting of sulfide-rich fluids directly from basalt orifices. Within 20 months past the eruption, maximal temperatures in the hot zone had dropped from 30°C to below 15°C, and sulfide concentration had declined substantially compared to the post-eruption maximum. However, spatial variation in environmental conditions across the hot, warm, and cool zones persisted, and supported different faunas across those zones. This environmental variation across the vent landscape creates a mosaic of habitat types, with variable chemical resource supply and abiotic constraints, which needs to be considered when characterizing succession and quantifying recovery.

After two years into the successional sequence, new species continued to arrive (10 total), even though the temperatures had stabilized to a range of 2–10°C across the habitats, and to 2.0–4.2° C directly at the sandwiches. Other species disappeared as colonists (11 total), although they may have persisted in the surrounding community as adults. In these later years, it is likely that differences in species' life-history traits, such as reproduction and larval dispersal [29], and species interactions [22] become more important in driving patterns such as the gradual increase in species richness. These types of biological processes can be detected only through long-term monitoring and sampling across environmental zones.

Evaluating community recovery typically involves a comparison of species diversity or composition to a pre-disturbance, or undisturbed, 'baseline'. If we adopt this approach, using the pre-eruption species richness as baseline, we would conclude from rarefaction curves that P-vent was approaching recovery as early as 2 years after the eruption in the warm and cool habitats. In our observations, the diversity even surpassed the pre-eruption state after roughly 3 years. If we use species composition instead (occurrence and relative abundance in nMDS plots), we would conclude that the community was approaching recovery to baseline within 8–9 years after the eruption. However, the species composition was still changing at 11 years, and was dissimilar to the pre-eruption state, indicating that the communities may have bypassed the pre-eruption communities along the successional sequence, and may continue to change. The pre-eruption observations likely did not represent an equilibrium or climax community, as they were from samples collected 7 years after the previous eruption in 1991. This comparison illustrates the challenge in selecting and evaluating a pre-disturbance baseline community. An alternative approach to quantify recovery is to compare community composition during succession to regional species pools. Using the Desbruyères *et al.* [27] species list for 9° N EPR, we found only 42% of the regional species pool in our pre-eruption samples, 73% across all of the post-eruption samples, and only 32% at the final 11-year date, suggesting that further changes over time may occur.

Prior studies of vent community recovery provide examples of how undersampling the diversity of habitat type or the species in a baseline community can lead to an underestimate of recovery time at a disturbed site. At Juan de Fuca Ridge, Tunnicliffe *et al.* [12] reported that 16 species out of a baseline pool of 30 (53%) had re-established within 2 years of an eruption, plus an additional 10 species not in the baseline. The baseline had been documented at one time point from a single, nearby, undisturbed vent with limited habitat diversity, and so did not represent the full diversity of potential source colonists in the area, leading to an underestimate of recovery time. On the EPR after the 1991 eruption near 9° N, Shank *et al.* [9] observed 32 species out of a baseline pool of 45 (71%) had returned by 4.5 years after the disturbance but faunal lists did not include many small species that are typical of vent communities at EPR [27]. Small species tend to arrive later in vent successional sequences [13,25], so a sampling method that misses them will likely result in a faster estimate of recovery time.

Interpretation of field observations of vent community recovery from disturbance differs substantially depending on whether the focus is on re-assembly of species at a single vent field, or on the persistence of species diversity across sites in a region (i.e. across the metacommunity). From the single-site perspective, the question is simply how quickly the pre-disturbance community becomes established, and it does not matter whether the pre-eruption state had low species diversity due to recent prior disturbance or to low habitat diversity. From a metacommunity perspective, it becomes important to know whether a sufficient number of communities, occupying a spectrum of habitat diversity, recover from disturbance far enough through the successional stages to support regional species diversity. If disturbance becomes too frequent, or spatially extensive, to support a mosaic of communities representing the full spectrum of successional stages, then diversity of the regional species pool will decline. This more complicated perspective requires sampling at multiple sites, but is critical for understanding the dynamics and distributions of vent species.

Such an understanding is precisely what policymakers are calling for in order to anticipate and manage the impacts of mining of vent deposits [20] and other seafloor resources [30,31]. Our observations of prolonged recovery time at a frequently disturbed vent site, previously expected to recover particularly rapidly, caution that vent communities, in general,

may be less resilient to mining disturbance than originally expected. This reinforces the importance of assessing community recovery against baselines that account for regional factors such as habitat spacing, connectivity, and regional species pools. Mining will take place over expansive areas for prolonged periods of time, which poses particular risk to species with low colonization rates when many adjacent sites are destroyed. Importantly, an understanding of frequently disturbed vent communities in the eastern Pacific, even from the metacommunity perspective, cannot be extrapolated to predict recovery from mining of seafloor sulfides elsewhere in the world's oceans. This study nevertheless suggests that multiple factors can influence the dynamics of re-establishing communities in naturally unstable systems. This is particularly critical as species diversity has been described only for a few vent sites, and we know even less about the diversity of habitat types and their temporal stability, particularly on complex arc and back-arc systems [32]. Furthermore, sites of particular interest for exploitation include large sulfide deposits that appear to have been active for thousands of years [33], and may be located far from other potential source communities, leading to very different recovery dynamics.

Data accessibility. Data on colonists, habitat, and environmental conditions are available in the publicly accessible BCO-DMO database (https://www.bco-dmo.org/) as doi:10.26008/1912/bco-dmo. 733173.2 and doi:10.26008/1912/bco-dmo.733210.3.

Authors' contributions. L.S.M. designed the colonization experiment, analysed the faunal data, and wrote the initial manuscript draft; S.W.M. conducted fieldwork and led the species identification; N.L.B. measured and analysed fluid chemistry; S.E.B. contributed to field and laboratory studies and data analyses, S.M.S. conducted fieldwork; L.N.D. contributed faunal data. All authors participated in manuscript preparation.

Competing interests. We declare we have no competing interests.

Funding. Support was provided by NSF grant nos. OCE-1356738, DEB-1558904 and OCE-1829773 to L.S.M., and NSF grant nos. OCE-0452333, OCE-1136727, OCE-1131095, and OCE-1559198 to S.M.S. Support from Ifremer 'Geobiology of Extreme Environment', EU ITN SENSENET no. 237868, CNRS INEE, and Fondation Total was provided to N.L.B., and from the French Oceanographic Research Fleet, CNRS and Sorbonne University for the MESCAL cruise (doi:10.17600/12010020).

Acknowledgements. We appreciate the contributions of the Alvin and Nautile groups, the Captain and crew of Atlantis and L'Atalante, and Chief Scientists Karen von Damm, Andreas Thurnherr, and Jim Ledwell on the RESET, LADDER-I and LADDER-II cruises, respectively. Tim Shank and Breea Govenar generously set out the initial colonization experiments in June 2006. We thank J.P. Brulport for the preparation of chemical sensors. Insightful comments by two anonymous reviewers improved the manuscript. We dedicate this paper to honour the memory of Diane Poehls Adams who was pivotal in launching this study.

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
