## [Reviewer comments · Proceedings of the Royal Society B: Biological Sciences]

Review History

RSPB-2020-2070.R0 (Original submission)

Review form: Reviewer 1

Recommendation

Accept with minor revision (please list in comments)

Scientific importance: Is the manuscript an original and important contribution to its field?

Excellent

General interest: Is the paper of sufficient general interest?

Excellent

Quality of the paper: Is the overall quality of the paper suitable?

Excellent

Is the length of the paper justified?

Yes

Should the paper be seen by a specialist statistical reviewer?

No

Do you have any concerns about statistical analyses in this paper? If so, please specify them explicitly in your report.

Yes

It is a condition of publication that authors make their supporting data, code and materials available - either as supplementary material or hosted in an external repository. Please rate, if applicable, the supporting data on the following criteria.

Is it accessible?

No

Is it clear?

Yes

Is it adequate?

Yes

Do you have any ethical concerns with this paper?

No

Comments to the Author

The MS showed the long-term oscillation of the faunal community structure in deep-sea hydrothermal vent field where faces mining impact in the global seafloor. The MS provides very important knowledge that the community does not reach a plateau or recover within a few years after a disturbance that has been known from 1990s, but keeps changing, based on the long-term monitoring after natural disturbance in the eastern Pacific hydrothermal vent fields. Their findings are not only scientifically but socially or politically important to elucidate deep-sea ecosystem, which occupies more than 90% of the earth's biosphere. We need to re-think the meanings of the word "resilience", under the metacommunity perspectives.

I agreed with the advocates of the MS and I would encourage the authors to process their datasets a bit more to be understood easily. For example, just to plot environmental parameters on nMDS may help the reader to understand the contribution of temperature or duration after the eruption to the community composition.

Although it is not based on the direct observation, a model also predicts that it takes more than 40 years to recover 70% of the community, in the western Pacific. The following paper may help the discussion by the authors;

Suzuki, K., Yoshida, K., Watanabe, H., & Yamamoto, H. (2018). Mapping the resilience of chemosynthetic communities in hydrothermal vent fields. *Scientific reports*, 8(1), 1-8.

Some detailed comments are follows;

Line 148 - 155: Although nMDS provides easy-to-understand graphics, the placement of each community may sometimes be distorted in 2-D ordination. Similarity index for each paired community may help the reader to confirm that the nMDS plot represents the similarity of faunal composition, and I would suggest the authors to provide the similarity index matrix, as a Supplemental data.

Line 155 - 156: I expect that there are benthic copepods or other multi-cellular meiofauna on the sandwiches but they are not found in Supplementary Data Table 1. Were they not there? Or they escaped during the recovery of the sandwiches, or removed from the analyses?

Line 187 - 188: I found the dataset at the following URLs;
<https://www.bco-dmo.org/dataset/733210/data>

<https://www.bco-dmo.org/dataset/733173/data>

It may be better to show the direct link of the datasets, instead of showing the home of the database.

Line 271: If the authors would like to focus on “evenness” of the community, it would be better to help to discuss more details by showing bar graphs of species composition of each sandwich and/or evenness index (e.g. Pielou’s J’) with species richness, or just the diversity index that takes into account both species richness and evenness (e.g. Shannon index).

Figure 2: A few unexpected lines were placed around Fig. 2a.

Supplementary Data Table 1: “Date” may be "Month", as in the main text.

Review form: Reviewer 2

Recommendation

Accept as is

Scientific importance: Is the manuscript an original and important contribution to its field?

Excellent

General interest: Is the paper of sufficient general interest?

Acceptable

Quality of the paper: Is the overall quality of the paper suitable?

Excellent

Is the length of the paper justified?

Yes

Should the paper be seen by a specialist statistical reviewer?

No

Do you have any concerns about statistical analyses in this paper? If so, please specify them explicitly in your report.

No

It is a condition of publication that authors make their supporting data, code and materials available - either as supplementary material or hosted in an external repository. Please rate, if applicable, the supporting data on the following criteria.

Is it accessible?

Yes

Is it clear?

Yes

Is it adequate?

Yes

Do you have any ethical concerns with this paper?

No

Comments to the Author

In "Prolonged recovery time after eruptive disturbance of a deep-sea hydrothermal vent community" the authors provide an unprecedented and unparalleled analysis of recovery and community succession of a hydrothermal vent ecosystem following catastrophic natural disturbance. This is an important and timely study as the deep-sea mining industry advances towards production. There are no major methodological or analytical issues with this paper. This study is representative of best-in-class research on disturbance and resilience at deep-sea hydrothermal vents and will be an important addition to the scientific literature. It is acceptable without any revisions.

I have only a few small notes: The rarefaction curves on figure 4: e and f are difficult to read as they are forced into the same scale as 4: d. A series of papers led by Sen and Fisher on community succession at Lau Basin hydrothermal vents that shows relative stability (i.e. <https://aslopubs.onlinelibrary.wiley.com/doi/abs/10.4319/lo.2014.59.5.1510>) might serve as a useful reference to include in the discussion.

Decision letter (RSPB-2020-2070.R0)

28-Sep-2020

Dear Dr Mullineaux:

Your manuscript has now been peer reviewed and the reviews have been assessed by an Associate Editor. The reviewers' comments (not including confidential comments to the Editor) and the comments from the Associate Editor are included at the end of this email for your reference. As you will see, the reviewers and the Editors have raised some concerns with your manuscript and we would like to invite you to revise your manuscript to address them. One of the concerns was the availability of the data. Acceptance of your ms will rely on the data being placed in an appropriate data repository.

Research ethics:

Use of animals and field studies:

It is a condition of publication that you make available the data and research materials supporting the results in the article. Please see our Data Sharing Policies (<https://royalsociety.org/journals/authors/author-guidelines/#data>). Datasets should be deposited in an appropriate publicly available repository and details of the associated accession number, link or DOI to the datasets must be included in the Data Accessibility section of the article (<https://royalsociety.org/journals/ethics-policies/data-sharing-mining/>). Reference(s) to datasets should also be included in the reference list of the article with DOIs (where available).

Please submit a copy of your revised paper within three weeks. If we do not hear from you within this time your manuscript will be rejected. If you are unable to meet this deadline please let us know as soon as possible, as we may be able to grant a short extension.

Best wishes,
 Dr Daniel Costa
 mailto: proceedingsb@royalsociety.org

Reviewer(s)' Comments to Author:

Referee: 1

Comments to the Author(s)

The MS showed the long-term oscillation of the faunal community structure in deep-sea hydrothermal vent field where faces mining impact in the global seafloor. The MS provides very important knowledge that the community does not reach a plateau or recover within a few years after a disturbance that has been known from 1990s, but keeps changing, based on the long-term monitoring after natural disturbance in the eastern Pacific hydrothermal vent fields. Their findings are not only scientifically but socially or politically important to elucidate deep-sea ecosystem, which occupies more than 90% of the earth's biosphere. We need to re-think the meanings of the word "resilience", under the metacommunity perspectives.

I agreed with the advocates of the MS and I would encourage the authors to process their datasets a bit more to be understood easily. For example, just to plot environmental parameters on nMDS may help the reader to understand the contribution of temperature or duration after the eruption to the community composition.

Although it is not based on the direct observation, a model also predicts that it takes more than 40 years to recover 70% of the community, in the western Pacific. The following paper may help the discussion by the authors;

Suzuki, K., Yoshida, K., Watanabe, H., & Yamamoto, H. (2018). Mapping the resilience of chemosynthetic communities in hydrothermal vent fields. *Scientific reports*, 8(1), 1-8.

Some detailed comments are follows;

Line 148 – 155: Although nMDS provides easy-to-understand graphics, the placement of each community may sometimes be distorted in 2-D ordination. Similarity index for each paired community may help the reader to confirm that the nMDS plot represents the similarity of faunal composition, and I would suggest the authors to provide the similarity index matrix, as a Supplemental data.

Line 155 – 156: I expect that there are benthic copepods or other multi-cellular meiofauna on the sandwiches but they are not found in Supplementary Data Table 1. Were they not there? Or they escaped during the recovery of the sandwiches, or removed from the analyses?

Line 187 – 188: I found the dataset at the following URLs;

<https://www.bco-dmo.org/dataset/733210/data>

<https://www.bco-dmo.org/dataset/733173/data>

It may be better to show the direct link of the datasets, instead of showing the home of the database.

Line 271: If the authors would like to focus on "evenness" of the community, it would be better to help to discuss more details by showing bar graphs of species composition of each sandwich and/or evenness index (e.g. Pielou's J') with species richness, or just the diversity index that takes into account both species richness and evenness (e.g. Shannon index).

Figure 2: A few unexpected lines were placed around Fig. 2a.

Supplementary Data Table 1: "Date" may be "Month", as in the main text.

Referee: 2

Comments to the Author(s)

In "Prolonged recovery time after eruptive disturbance of a deep-sea hydrothermal vent community" the authors provide an unprecedented and unparalleled analysis of recovery and community succession of a hydrothermal vent ecosystem following catastrophic natural disturbance. This is an important and timely study as the deep-sea mining industry advances towards production. There are no major methodological or analytical issues with this paper. This study is representative of best-in-class research on disturbance and resilience at deep-sea hydrothermal vents and will be an important addition to the scientific literature. It is acceptable without any revisions.

I have only a few small notes: The rarefaction curves on figure 4: e and f are difficult to read as they are forced into the same scale as 4: d. A series of papers led by Sen and Fisher on community succession at Lau Basin hydrothermal vents that shows relative stability (i.e. <https://aslopubs.onlinelibrary.wiley.com/doi/abs/10.4319/lo.2014.59.5.1510>) might serve as a useful reference to include in the discussion.

Author's Response to Decision Letter for (RSPB-2020-2070.R0)

See Appendix A.

Decision letter (RSPB-2020-2070.R1)

22-Nov-2020

Dear Dr Mullineaux

I am pleased to inform you that your Review manuscript RSPB-2020-2070.R1 entitled "Prolonged recovery time after eruptive disturbance of a deep-sea hydrothermal vent community" has been accepted for publication in Proceedings B.

The referee(s) do not recommend any further changes. Therefore, please proof-read your manuscript carefully and upload your final files for publication. Because the schedule for publication is very tight, it is a condition of publication that you submit the revised version of your manuscript within 7 days. If you do not think you will be able to meet this date please let me know immediately.

To upload your manuscript, log into <http://mc.manuscriptcentral.com/prsb> and enter your Author Centre, where you will find your manuscript title listed under "Manuscripts with Decisions." Under "Actions," click on "Create a Revision." Your manuscript number has been appended to denote a revision.

You will be unable to make your revisions on the originally submitted version of the manuscript. Instead, upload a new version through your Author Centre.

- 1) A text file of the manuscript (doc, txt, rtf or tex), including the references, tables (including captions) and figure captions. Please remove any tracked changes from the text before submission. PDF files are not an accepted format for the "Main Document".
- 2) A separate electronic file of each figure (tiff, EPS or print-quality PDF preferred). The format should be produced directly from original creation package, or original software format. Please note that PowerPoint files are not accepted.
- 3) Electronic supplementary material: this should be contained in a separate file from the main text and the file name should contain the author's name and journal name, e.g. `authorname_procb_ESM_figures.pdf`
All supplementary materials accompanying an accepted article will be treated as in their final form. They will be published alongside the paper on the journal website and posted on the online figshare repository. Files on figshare will be made available approximately one week before the accompanying article so that the supplementary material can be attributed a unique DOI. Please see: <https://royalsociety.org/journals/authors/author-guidelines/>

4) Data-Sharing and data citation

It is a condition of publication that data supporting your paper are made available. Data should be made available either in the electronic supplementary material or through an appropriate repository. Details of how to access data should be included in your paper. Please see <https://royalsociety.org/journals/ethics-policies/data-sharing-mining/> for more details.

If you wish to submit your data to Dryad (<http://datadryad.org/>) and have not already done so you can submit your data via this link <http://datadryad.org/submit?journalID=RSPB&manu=RSPB-2020-2070.R1> which will take you to your unique entry in the Dryad repository.

Once again, thank you for submitting your manuscript to Proceedings B and I look forward to receiving your final version. If you have any questions at all, please do not hesitate to get in touch.

Sincerely,
Dr Daniel Costa
Editor, Proceedings B
<mailto:proceedingsb@royalsociety.org>

Associate Editor Board Member: 1
Comments to Author:

The authors have incorporated the comments of the reviewers into the revised ms and all the concerns raised have been addressed. I would add two minor caveats, which can be dealt with at the proof stage: first, that there are references cited in the abstract, which is not journal style. Second, that the abstract should be revised to reflect and emphasise that the results of this study apply to EPR and may not be extendable to hydrothermal vents in general (slower spreading ridges may be even slower to recover!). This is a welcome and timely contribution to PRSB.

Julia Sigwart

Decision letter (RSPB-2020-2070.R2)

27-Nov-2020

Dear Dr Mullineaux

I am pleased to inform you that your manuscript entitled "Prolonged recovery time after eruptive disturbance of a deep-sea hydrothermal vent community" has been accepted for publication in Proceedings B.

Your article has been estimated as being 8 pages long. Our Production Office will be able to confirm the exact length at proof stage.

Open Access

Paper charges

Sincerely,

Appendix A

Lauren S. Mullineaux, *Senior Scientist*

Biology Department, MS# 38

266 Woods Hole Road, Woods Hole, MA 02543

Office: 508 289-2898 | lmullineaux@whoi.edu

web.whoi.edu/mullineaux

26 October 2020

Dan Costa
Associate Editor, *Proceedings of the Royal Society B*

Dear Dr. Costa,

My coauthors and I are pleased to submit this revision of our manuscript 'Prolonged recovery after eruptive disturbance of a deep-sea hydrothermal vent' to *Proceedings of the Royal Society B*. The reviews were very helpful and we responded to all the comments (details below).

Thank you for your consideration,

Lauren Mullineaux
Senior Scientist

Responses to Referees:

Referee: 1

I would encourage the authors to process their datasets a bit more to be understood easily.

Fig 4: We decided to plot temperature on the nMDS, as it was the one environmental measurement directly coupled to each colonization surface

A model also predicts that it takes more than 40 years to recover 70% of the community, in the western Pacific.

Line 80 and References: We now cite Suzuki et al (2018) in the Introduction

Although nMDS provides easy-to-understand graphics, the placement of each community may sometimes be distorted in 2-D ordination.

Table 3: We include a Pearson's correlation matrix for each nMDS plot as Supplementary Data

I expect that there are benthic copepods or other multi-cellular meiofauna on the sandwiches but they are not found in Supplementary Data Table 1.

Line 145: Meiofauna were indeed found on the sandwiches, but were not included in analyses because they were not reliably quantified by our techniques – e.g., they were too mobile or small to recover reliably, or were outside our taxonomic expertise.

Show the direct link of the datasets, instead of showing the home of the database.

Line 190: The datasets are now published and we include the DOIs in the text

If the authors would like to focus on “evenness” of the community, it would be better to help to discuss more details ... (e.g. Shannon index).

Line 274: We realize this section was unclear and revised to emphasize that over the course of succession, species had replaced each other, so even though diversity did not increase after ~2 years, the composition of the community continued to change. We calculated Shannon H for pooled data from each recovery date and zone, but decided not to include because it provided no additional insight beyond the rarefaction curves.

A few unexpected lines were placed around Fig. 2a.

Fig 2: We spruced up panel alignment and line widths in Fig. 2 a, and removed extra lines

Supplementary Data Table 1: “Date” may be “Month”, as in the main text.

Table 1: We changed the Column label to ‘Month’

Referee: 2

The rarefaction curves on figure 4: e and f are difficult to read as they are forced into the same scale as 4: d.

Fig. 4: We need to keep the horizontal scale consistent across panels 4 e, d, and f, but have now stretched the horizontal axis of all three to make the patterns in e and f more visible

A series of papers led by Sen and Fisher on community succession at Lau Basin hydrothermal vents that shows relative stability might serve as a useful reference

Line 82 and References: We agree. We found a similar example for MAR (Copley et al. 2007) and now cite both that paper and Sen et al 2014